# The roles of history, chance, and natural selection in the evolution of antibiotic resistance

**Alfonso Santos-Lopez[1]\*[†‡], Christopher W Marshall[1]\*[†§], Allison L Haas[1], Caroline Turner[1#], Javier Rasero[2], Vaughn S Cooper[1,3]\***

[1]Department of Microbiology and Molecular Genetics, School of Medicine, University of Pittsburgh, Pittsburgh, United States; [2]Department of Psychology, Carnegie Mellon University, Pittsburgh, United States; [3]Center for Evolutionary Biology and Medicine, University of Pittsburgh, Pittsburgh, United States

**\*For correspondence:**
alfonsosantos2@hotmail.com
(AS-L);
christopher.marshall@marquette.
edu (CWM);
vaughn.cooper@pitt.edu (VSC)

[†]These authors contributed
equally to this work

**Present address:** [‡]Servicio
de Microbiología. Hospital
Universitario Ramón y Cajal
and Instituto Ramón y Cajal de
Investigación Sanitaria, Madrid,
Spain; [§]Department of Biological
Sciences, Marquette University,
Milwaukee, United States;
[#]Department of Biology, Loyola
University Chicago, Chicago,
United States

**Competing interest:** The authors
declare that no competing
interests exist.

**Reviewing Editor:** María
Mercedes Zambrano, CorpoGen,
Colombia

**Abstract** History, chance, and selection are the fundamental factors that drive and constrain evolution. We designed evolution experiments to disentangle and quantify effects of these forces on the evolution of antibiotic resistance. Previously, we showed that selection of the pathogen *Acinetobacter baumannii* in both structured and unstructured environments containing the antibiotic ciprofloxacin produced distinct genotypes and phenotypes, with lower resistance in biofilms as well as collateral sensitivity to β-lactam drugs (Santos-Lopez et al., 2019). Here we study how this prior history influences subsequent evolution in new β-lactam antibiotics. Selection was imposed by increasing concentrations of ceftazidime and imipenem and chance differences arose as random mutations among replicate populations. The effects of history were reduced by increasingly strong selection in new drugs, but not erased, at times revealing important contingencies. A history of selection in structured environments constrained resistance to new drugs and led to frequent loss of resistance to the initial drug by genetic reversions and not compensatory mutations. This research demonstrates that despite strong selective pressures of antibiotics leading to genetic parallelism, history can etch potential vulnerabilities to orthogonal drugs.

## Introduction

Evolution can be propelled by natural selection, it can wander with the chance effects of mutation and genetic drift, and it can be constrained by history, whereby past events limit or even potentiate the future (*Travisano et al., 1995*; *Keller and Taylor, 2008*; *Meyer et al., 2012*; *Kryazhimskiy et al., 2014*; *Rebolleda-Gomez and Travisano, 2019*). The relative roles of these forces has been debated, with the constraints of history the most contentious (*Blount et al., 2018*). A wealth of recent research has shown that evolution can be surprisingly repeatable when selection is strong even among distantly related lineages or in different environments (*Lieberman et al., 2011*; *Lassig et al., 2017*; *Turner et al., 2018*), but disparate outcomes become more likely as the footprint of history (i.e. differences in genetic background caused by chance and selection in different environments) increases (*Blount et al., 2018*; *Benton et al., 2021*; *Mahrt et al., 2021*) (For definitions of the forces and their role in the evolution of antibiotic resistance, see *Box 1*). In the absence of chance and history, selection will cause the most fit genotype to fix in the particular environment, and provided this variant is available, evolution will be perfectly predictable (*Bailey et al., 2015*; *Lassig et al., 2017*). However, historical and stochastic processes inevitably produce some degree of contingency, making evolution less predictable, reflecting the importance of evolutionary history (*Blount et al., 2008*; *Meyer et al., 2012*; *Bajić et al., 2018*; *Blount et al., 2018*; *Card et al., 2019*; *Galardini et al., 2019*). The evolution

## Box 1. Definitions of selection, chance, and history in the evolution of AMR.

Antibiotics impose strong selective pressures on microbial populations, which can produce highly repeatable outcomes when bacterial population sizes are large and mutations are not limiting. In the absence of chance and history, **selection**, the process by which heritable traits that increase survival and reproduction rise in population frequency, will cause the fixation of the resistant allele associated with the highest fitness in the population, making evolution highly predictable. However, the origin of genetic variation is a stochastic process. **Chance** effects of acquiring a mutation, gene, or mobile element, or changes in the frequencies of these alleles by genetic drift determine whether, by what mechanism, and to what degree, resistance evolves in a given population. Furthermore, the evolutionary **history** of a population can produce contingencies that can make evolution unpredictable. For instance, different genetic backgrounds shaped in different environments can alter the phenotype of a given mutation. History can therefore alter the occurrence, mechanism, degree, and success of antimicrobial resistance.

Antibiotic treatments usually target advanced infections, which implies medium to large bacterial population sizes (*Palaci et al., 2007*). Estimates suggest that a typical antibiotic treatment above the MIC concentration will clear the infection with a probability higher than 99 % (*Paterson et al., 2016*). But some bacterial infections can be established from as few as 10 cells (*Jones et al., 2016*), so even small surviving subpopulations could re-infect the host. Thus, we might expect that strong selection imposed by antibiotics acting on large populations would be powerful enough to overwhelm the constraints of history. The large population sizes also might enable many mutations to be accessible in each infection, which would diminish the effects of chance. However, bottlenecks produced by the antibiotic could increase effects of drift and amplify contributions of chance and history. By propagating large populations under sequential bottlenecks, we can reproduce some of the population dynamics of the establishment and clearance of infections, and by applying the framework of *Travisano et al., 1995*, we can quantify the roles of history, chance, and selection in adaptation to antibiotics.

of a new trait, whether by horizontally acquired genes or de novo mutation, is a stochastic process that depends on available genetic variation capable of producing a new trait (*Khan et al., 2011*; *Salverda et al., 2011*). As any other evolved trait, antimicrobial resistance (AMR) is subjected to these three evolutionary forces (*Box 1*).

Antibiotics can impose strong selection pressure on microbial populations, leading to highly repeatable evolutionary outcomes (*Vogwill et al., 2014*; *Lukačišinová et al., 2020*; *Scribner et al., 2020*), with the level of parallelism predicted to depend on the strength of antibiotic pressure (*Wistrand-Yuen et al., 2018*). However, evolutionary history can also alter the distribution of fitness effects of AMR mutations, their mechanisms of action, or their degree of conferred resistance (*Eyre-Walker and Keightley, 2007*; *Hall and MacLean, 2011*; *Yen and Papin, 2017*; *Barbosa et al., 2019*). The phenotypic effect of any given mutation acquired is contingent on prior events and will determine the potential of further adaptations to a given environment (*Travisano et al., 1995*). For example, the effects of a given mutation can vary in different genetic backgrounds (epistasis) or in different environments (pleiotropy) and those mutations can constrain further adaptations (*Trindade et al., 2009*; *Hall and MacLean, 2011*; *Yen and Papin, 2017*; *Gifford et al., 2018*; *Santos-Lopez et al., 2019*). Additionally, chance differences in the mutations acquired, their order of occurrence, or compensatory mutations that decrease resistance costs can affect the eventual level of resistance and its evolutionary success in the population (*Salverda et al., 2011*; *Wistrand-Yuen et al., 2018*).

The study of mutational pathways to AMR has become accessible by applying population-wide whole-genome sequencing (WGS) to experimentally evolved populations (for a review, see *Baquero, 2021*). Growth in antibiotics will select for resistant phenotypes whose genotypes can be determined

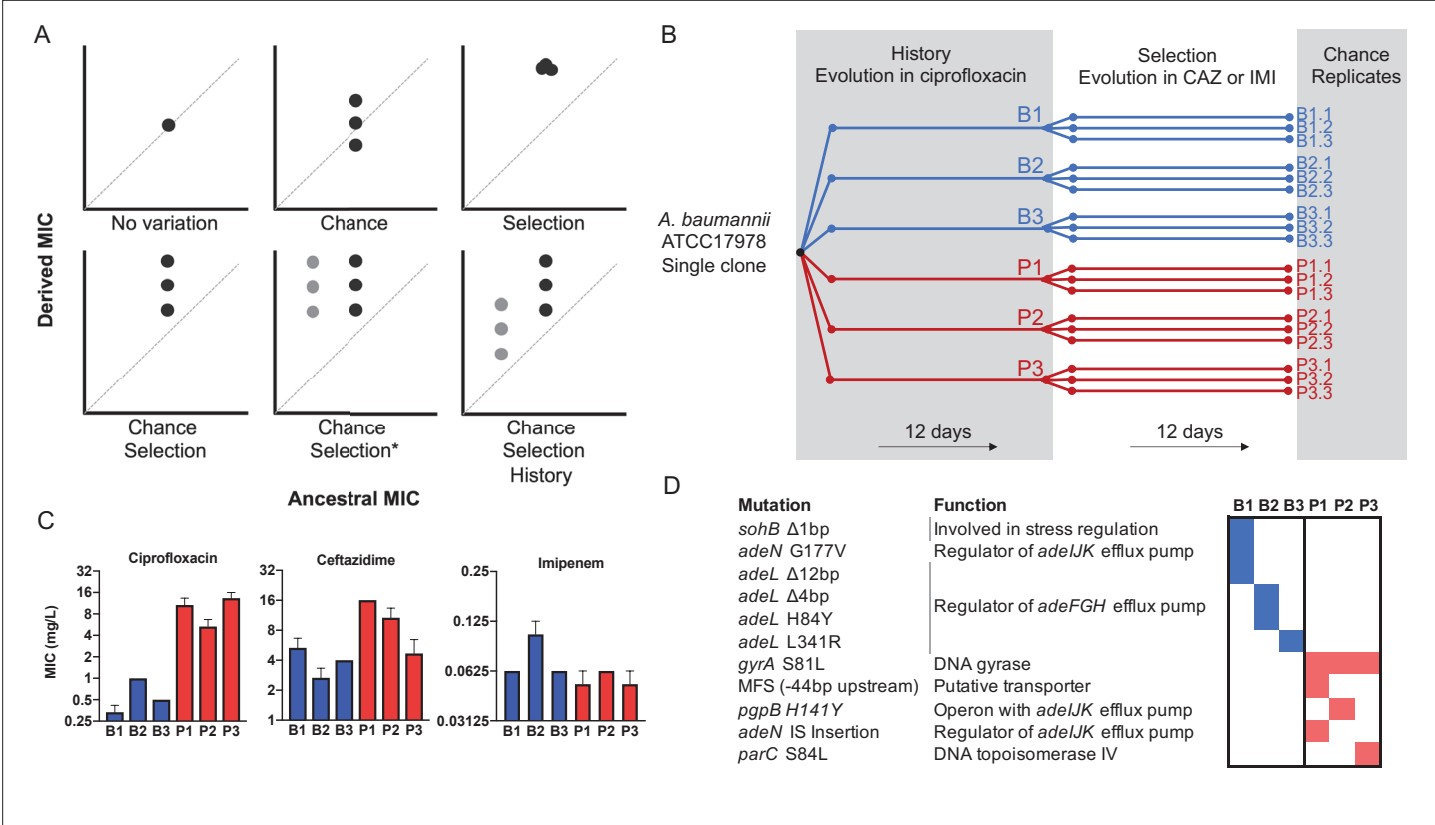

**Figure 1.** Experimental design to differentiate history, chance, and selection including starting genotypes and AMR phenotypes. (**A**) Potential outcomes of replicate evolved populations estimated by the resistance level before and after the antibiotic treatment. Grey and black symbols denote starting clones with different resistance levels. A more detailed description of this design is in the Methods section, modified from *Travisano et al., 1995*. The asterisk denotes the case in which chance and selection both erase historical effects. (**B**) Six different clones with distinct genotypes and CIP susceptibility were used to found new replicate populations that evolved in increasing CAZ or IMI for 12 days (*Santos-Lopez et al., 2019*). (**C**) MIC of the six ancestors in CIP, CAZ and IMI (± SEM). (**D**) Ancestral genotypes prior to the selection phase.

The online version of this article includes the following figure supplement(s) for figure 1:

**Source data 1.** Concentrations of CAZ and IMI (mg/L) added to the broth at different intervals of the evolution experiments.

**Source data 2.** Minimum inhibitory concentration (MIC) values for all ancestors and evolved clones by treatment.

**Figure supplement 1.** Resistance levels to ciprofloxacin, ceftazidime and imipenem of the ancestral strain prior to being propagated in the historical phase under increasing concentrations of CIP.

by WGS, and their frequencies and trajectories indicate relative genotype fitness. When large populations, $1 \times 10^7$ CFU/mL or higher, of bacteria are propagated, the probability that every base pair is mutated at least once approaches 99 % after ~80 generations (*Lynch et al., 2016*; *Santos-Lopez et al., 2019*). Yet chance still remains important because most mutations are initially rare and subject to genetic drift until they reach a critical frequency of establishment, when selection dominates their fate (*Heffernan and Wahl, 2002*; *Good et al., 2017*; *Cooper, 2018*). Furthermore, many mutations arise concurrently and those with higher fitness tend to exclude other alleles, known as clonal interference. Thus, the success of new mutations will be determined by their survival of drift, the chance that they co-occur with other fit mutants, and by their relative fitness, which is shaped by selection and history (*Nguyen Ba et al., 2019*).

The contributions of history, chance, and selection to evolution can be measured using an elegant experimental design (depicted in *Figure 1A*, *Box 1*, and described in detail in the Methods) introduced by *Travisano et al., 1995*, in which replicate populations are propagated from multiple ancestral strains with different evolutionary histories. This experimental design has been used to quantify effects of these forces and to predict evolution in prokaryotes, eukaryotes and even digital organisms (*Travisano et al., 1995*; *Flores-Moya et al., 2008*; *Keller and Taylor, 2008*; *Meyer et al., 2012*;

*Kryazhimskiy et al., 2014*; *Matos et al., 2015*; *Rebolleda-Gomez and Travisano, 2019*; *Bundy et al., 2021*), but has not been applied to calculate effects of these forces in the evolution of AMR, one of the most critical threats in modern medicine. Here we use this framework to measure the relative roles of history, chance, and selection in the evolution of AMR phenotypes and genotypes in the ESKAPE (*E*nterococcus faecium, *S*taphylococcus aureus, *K*lebsiella pneumoniae, *A*cinetobacter baumannii, *P*seudomonas aeruginosa, and *E*nterobacter spp.) pathogen *Acinetobacter baumannii*, a leading agent of multidrug-resistant infections worldwide and named as an urgent threat by the CDC (*CDC, 2019*). Quantifying contributions of these evolutionary forces is essential if we are ever to predict the evolution of drug resistance of pathogens, including HIV and malaria, and of various cancers (*Hughes and Andersson, 2015*; *Verlinden et al., 2016*; *MacLean and San Millan, 2019*; *Pokhriyal et al., 2019*; *Gerstung et al., 2020*).

## Results

Previously (*Santos-Lopez et al., 2019*), we propagated a single clone of *A. baumannii* (strain 17978-mff) for 12 days or 80 generations in increasing concentrations of the fluoroquinolone antibiotic ciprofloxacin (CIP). In that experiment, which established the history for the present study and is analogous to prior exposure in a clinical setting, three replicate populations each were propagated in biofilm conditions or planktonic conditions (hereafter B1–B3 and P1–P3 respectively, *Figure 1B*). These environments selected for different genetic pathways to CIP resistance and replicate populations also diverged by chance, which produced the genetic and phenotypic histories of the ancestral strains in the current study (*Figure 1C,D*, *Figure 1—source data 1*). Key historical differences include reduced ceftazidime (CAZ) resistance in B populations but increased CAZ resistance in P populations (*Figure 1C Santos-Lopez et al., 2019*).

In the current study, the 'selection' phase (*Figure 1B*) involved experimental evolution in increasing concentrations of the cephalosporin CAZ or the carbapenem imipenem (IMI) for 12 days via serial dilution of planktonic cultures. CAZ or IMI concentrations were doubled every three days (ca. 20 generations), starting with 0.5× minimum inhibitory concentration (MIC; *Figure 1—source data 1*) for each clone and finishing with 4 × MIC, where maximum killing has been observed with β-lactams antibiotics (*Nightingale, 1980*). Each population was therefore exposed to the same selective pressure during evolutionary rescue. In this study design (*Figure 1A*, Supplementary Text), the extent of increased resistance represents selection, effects of chance are the phenotypic variation among triplicate populations propagated from the same ancestor, and differences between populations derived from different ancestors quantifies effects of history (*Figure 1B*).

While the scale of this experiment could seem small, it is well suited for studying the evolution of resistance as 160 generations correspond to ca. 100 days, 15 days, or 170 days of growth in patients of *Escherichia coli*, *P. aeruginosa*, or *Salmonella enterica*, respectively (*Gibson et al., 2018*). In addition, the genetic contributions of chance, history, and selection were determined by sequencing whole populations to a mean site coverage of 358 (S.D. ± 106) bases at the end of the experiment.

### Contributions of evolutionary forces under antibiotic treatment

Antibiotic treatments are designed to achieve sufficient concentration in vivo to clear the infection and prevent the development of new resistant mutants. However, for several reasons including poor drug pharmacokinetics, poor drug distribution, or poor patient compliance, antibiotic concentrations are often below the MIC in body compartments (*Andersson and Hughes, 2014*). It is expected that as drug concentrations increase, the strength of selection relative to other forces also increases. We therefore analyzed resistance phenotypes of the whole population after 3 days of evolution under subinhibitory drug concentration and after 12 days of evolution in increasing drug levels that concluded at four times the MIC. We analyzed population-wide resistance instead of measures of single isolates because heterogeneity can determine the success or failure of an antibiotic treatment in clinical scenarios (*Sánchez-Romero and Casadesús, 2014*; *Dewachter et al., 2019*).

We estimated the role of each force as described by *Travisano et al., 1995*. Briefly, we estimated the effect of history as the square root of the variance among all propagated populations; the effect of chance as the square root of the variance between the replicates propagated from the same ancestor, and the effect of selection was calculated as the difference in grand mean of the

propagated replicates and their ancestors (see Materials and methods for details of this calculations). We estimated effects of these forces during propagation in two antibiotics, CAZ and IMI, and present results of each treatment sequentially. First, after 3 days of growth in subinhibitory concentrations of CAZ, history explained the largest variation in resistance phenotypes (61.7 % of variation, p<0.05), with 30.7 % for selection and only 7.6 % chance (*Figure 2A,E*, Materials and methods). As expected, CAZ resistance increased overall, but some individual populations did not differ significantly from their ancestor (populations P2, P3, *Figure 2A*). However, by day 12, following propagation in 4 × MIC CAZ, the amount of variation explained by selection increased to 47.8 % and effects of history dropped to 31.4 % (*Figure 2B,E*), indicating that strong selective pressures can diminish or erase the effects of history.

Previous studies have shown that other evolved traits such as fitness itself show declining adaptability: less fit populations adapt faster and to a greater extent than more fit populations when propagated under the same environmental conditions (*Wiser et al., 2013*; *Kryazhimskiy et al., 2014*), which would lead to reduced variance in fitness traits among populations. This homogeneity indeed emerged as prolonged CAZ selection overcame historical variation. Populations with lower initial MICs, which by necessity were exposed to lower concentrations of CAZ, increased their resistance level more than populations with higher MICs (*Figure 2—figure supplement 1*), implying weak selection for further resistance in populations exceeding the MIC threshold and hence declining rates of resistance gains. This finding also suggests that the level of evolved resistance converges and may be predictable (*Meyer et al., 2012*; *Kryazhimskiy et al., 2014*), but effects of genetic background remain (*Figure 2*). Strong antibiotic selection has the potential to overcome but do not entirely eliminate historical differences in resistance.

## Evolutionary trade-offs arise from past antibiotic selection

Evolutionary trade-offs occur when changes in a given gene or trait increase fitness in one environment but reduce fitness in another. For example, a history of adaptation to one antibiotic could alter resistance and subsequent evolution in the presence of a subsequent antibiotic. The phenomena of cross-resistance and collateral sensitivity are specific examples of pleiotropy, where the mechanism of resistance to the initial drug either directly increases or decreases resistance to other drugs, respectively (*Pal et al., 2015*). Additionally, the resistance mechanism could interact with other genes or alleles in the genome, a form of epistasis, and also promote or impede resistance evolution. We hypothesized that resistance mechanisms arising during selection in CAZ would alter resistance to other antibiotics both by genotype-independent (pleiotropy) and genotype-dependent (epistasis) mechanisms. Recall that during the history phase of the experiment (*Santos-Lopez et al., 2019*), populations propagated in increasing concentrations of CIP became from 4- to 200-fold more resistant to CIP (*Figure 1C*, *Santos-Lopez et al., 2019*). Some of these strains also became more resistant to CAZ (populations P1–P3), while others became more susceptible (populations B1 and B3, for more details, see *Santos-Lopez et al., 2019*), and given that these populations originated from the same ancestor, this variation in collateral resistance phenotypes is best explained by pleiotropy. In the current study, after 12 days evolving in the presence of CAZ, the grand mean of CIP resistance levels did not change, so history was the dominant force shaping the MIC to CIP (*Figure 3A*). However, if we analyze the P and the B populations independently, B populations became significantly more sensitive to CIP but the P populations did not (*Figure 3A*), showing that the emergence of collateral sensitivity may depend on prior selection in different environments. These results also indicate that CAZ resistance mechanisms interact with CIP resistance in potentially useful ways.

We also tested if evolving in the presence of CAZ-altered resistance to the carbapenem antibiotic IMI (*Figure 3B*). As CAZ and IMI are both β-lactam antibiotics and mutations in efflux pumps can alter resistance to both (*Lee et al., 2017*), we predicted selection in CAZ would also increase IMI resistance and further, that the contributions of each evolutionary force to IMI resistance would follow that measured for CAZ (*Figure 2B*). As expected, all 12 populations evolved in CAZ became more resistant to IMI (two-tailed nested t-test p<0.0001, t = 7.507, df = 34), and selection was the most important force (p<0.05), explaining almost 44.3 % of the variation, while history contributed 23.0 % and chance 32.2 % (*Figure 3E*).

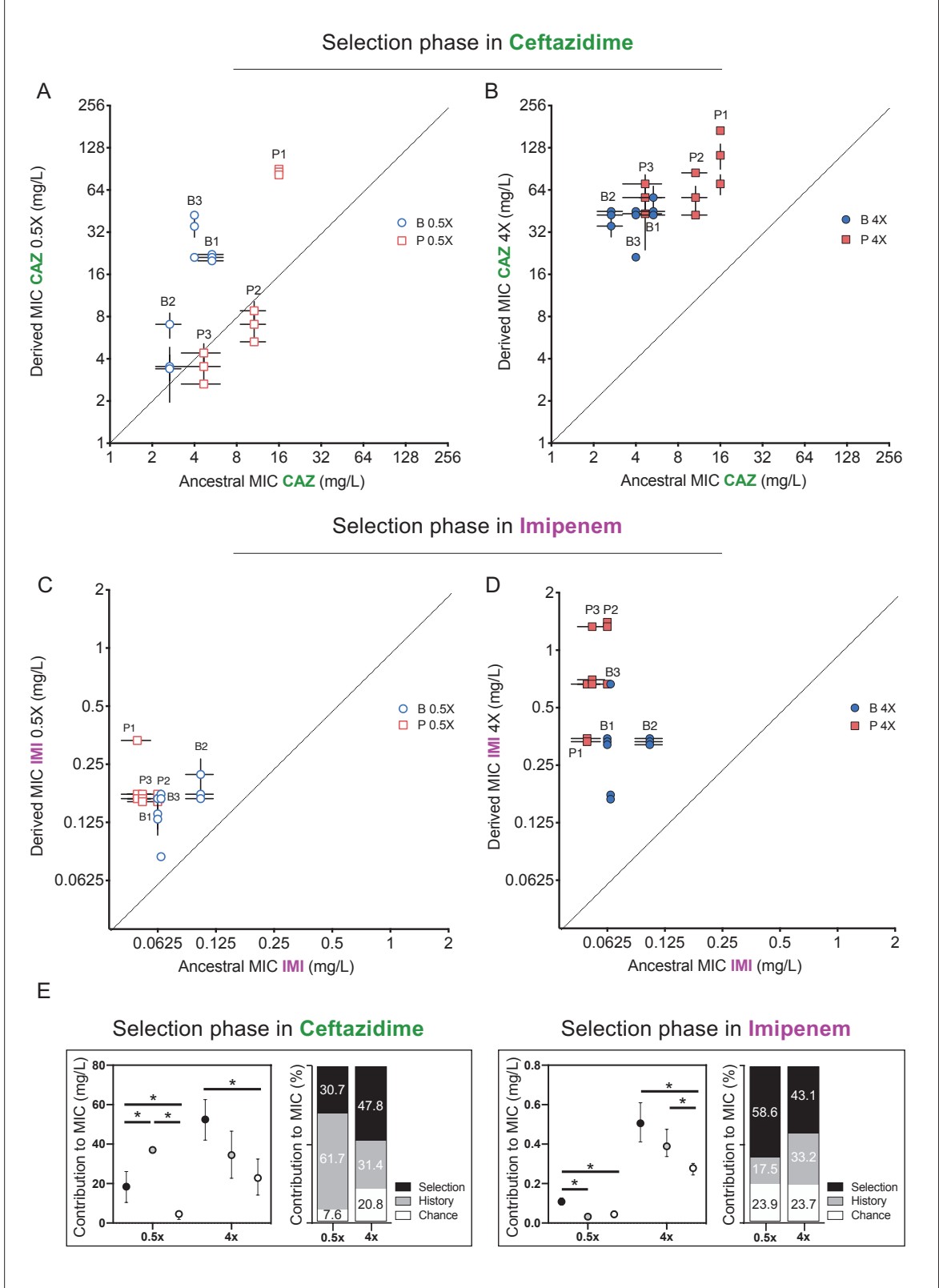

**Figure 2.** Effects of history, chance, and selection on the evolution of CAZ or IMI resistance after 3 days at 0.5 × MIC. (**A, C**) and after 12 days of increasing concentrations (**B, D**). Empty and filled symbols (3 days, left; and 12 days, right) represent CAZ or IMI MIC after 3 and 12 days of evolution. Blue symbols evolved from B ancestors were isolated from prior biofilm selection; red squares were evolved from P ancestors with a prior history in planktonic culture. Some symbols representing identical data points are jittered to be visible. MICs were measured in triplicate and shown± SEM. All

*Figure 2 continued on next page*

*Figure 2 continued*

populations increased CAZ resistance at day 3 (nested one-way ANOVA, Tukey's multiple comparison tests MIC day 0 vs. MIC day 3, p=0.0080 q = 4.428, df = 51) and at the end of the experiment (nested one-way ANOVA Tukey's multiple comparison tests MIC 0 vs. MIC day 12, p≤0.0001, q = 11.12, df = 51). All populations increased IMI resistance at day 12 but not at early timepoints (day 3) (nested one-way ANOVA Tukey's multiple comparison tests MIC at day 0 vs. MIC at day 12, p<0.0001, q = 9.519, df = 51; MIC at day 0 vs. MIC at day 3, p=0.3524, q = 1.969, df = 51). (**E**) Absolute and relative contributions of each evolutionary force. Error bars indicate 95 % confidence intervals. Asterisks denote p<0.05.

The online version of this article includes the following figure supplement(s) for figure 2:

**Source data 1.** Estimated statistics for history, chance, and selection forces.

**Figure supplement 1.** Correlation between ancestral MIC and increase of CAZ (top) and IMI (bottom) resistance after 3 and 12 days evolving in the presence of CAZ (left and right panels, respectively).

## Replaying the antibiotic treatment using a different antibiotic

We learned that the evolution of resistance in *A. baumannii* to one drug, CAZ, is substantially influenced by prior history of selection in another drug, CIP, as well as the prior growth environment, planktonic (P) or biofilm (B). Namely, B-derived populations evolved CAZ resistance at the expense of their prior CIP resistance, reversing this tradeoff. To test whether these results are repeatable and not limited to CAZ and CIP, we replayed the 'selection phase' with the same genotypes using the carbapenem IMI (*Figures 1 and 2 Santos-Lopez et al., 2019*). Here, no overall change in resistance occurred following 3 days in subinhibitory concentrations of IMI (*Figure 2C*) but did increase by experiment's end at 4 × MIC (*Figure 2D*). After the subinhibitory treatment, the more sensitive populations experienced greater gains in IMI resistance than the less sensitive populations, erasing some effects of history (*Figure 2C* and *Figure 2—figure supplement 1*). In total, selection again predominated (p<0.05) and explained 43.1 % of the phenotypic variation in this experiment, while history explained 33.2 % (*Figure 2B, D and F*).

As predicted by the CAZ experiment, evolution in IMI did not affect CIP resistance on average and history explained 75 % of the variation in MIC (*Figure 3F*), but again produced collateral sensitivity in two B populations (*Figure 3C*). This result demonstrates that mechanisms of IMI resistance also interact with historical resistance to CIP and produce tradeoffs. The biggest difference between the CAZ and IMI experiments is an asymmetry in cross-resistance between these drugs. Selection in CAZ increased IMI resistance (*Figure 3B*), but not *vice versa* (*Figure 3D*). These divergent cross-resistance networks result from the particular mutations that were selected in both experiments, which are explained below.

## Phenotypic divergence despite genetic parallelism

When multiple lineages evolve independently in the same environment, phenotypic convergence is usually observed, but the genotypes may be more variable (*Meyer et al., 2012*; *Bedhomme et al., 2013*; *Kryazhimskiy et al., 2014*). In our experiment, large populations were exposed to strong antibiotic pressure, so we predicted convergence at the genetic level owing to few solutions that improve both fitness and resistance (*Lenski, 2017*; *Cooper, 2018*). We conducted whole-population genomic sequencing of all populations at the end of the experiment to identify all contending mutations above a detection threshold of 5 % and analyzed the genetic contributions of history, chance, and selection using Manhattan distance estimators as a metric for the genotypic distance between populations (*Figure 4*). We calculated the genotypic role of chance as the mean distance between evolved populations sharing the same ancestor; history as the mean distance between evolved populations with different ancestors, after subtracting the effect of chance; and selection as the mean distance between ancestral and evolved populations, after subtracting the effects of chance and history. Using these metrics, we infer that evolution in CAZ at the genotypic level was shaped more by selection than history, but the opposite was seen in IMI, and effects of chance were similar in both experiments (*Figure 4*).

Clinical CAZ-resistant *A. baumannii* isolates commonly acquire mutations that increase the activity of <u>A</u>cinetobacter <u>d</u>rug <u>e</u>fflux (*ade*) pumps (*Lee et al., 2017*). In the history phase of CIP selection, biofilm lines (*Figure 1B*) selected mutations in *adeL*, the regulator of the *adeFGH* pump, which produce collateral sensitivity to CAZ and other β-lactams (*Figure 1D*). In contrast, P lines became cross-resistant to CAZ by *adeN* mutations that regulate the *adeIJK* complex or *pgpB* mutations that are

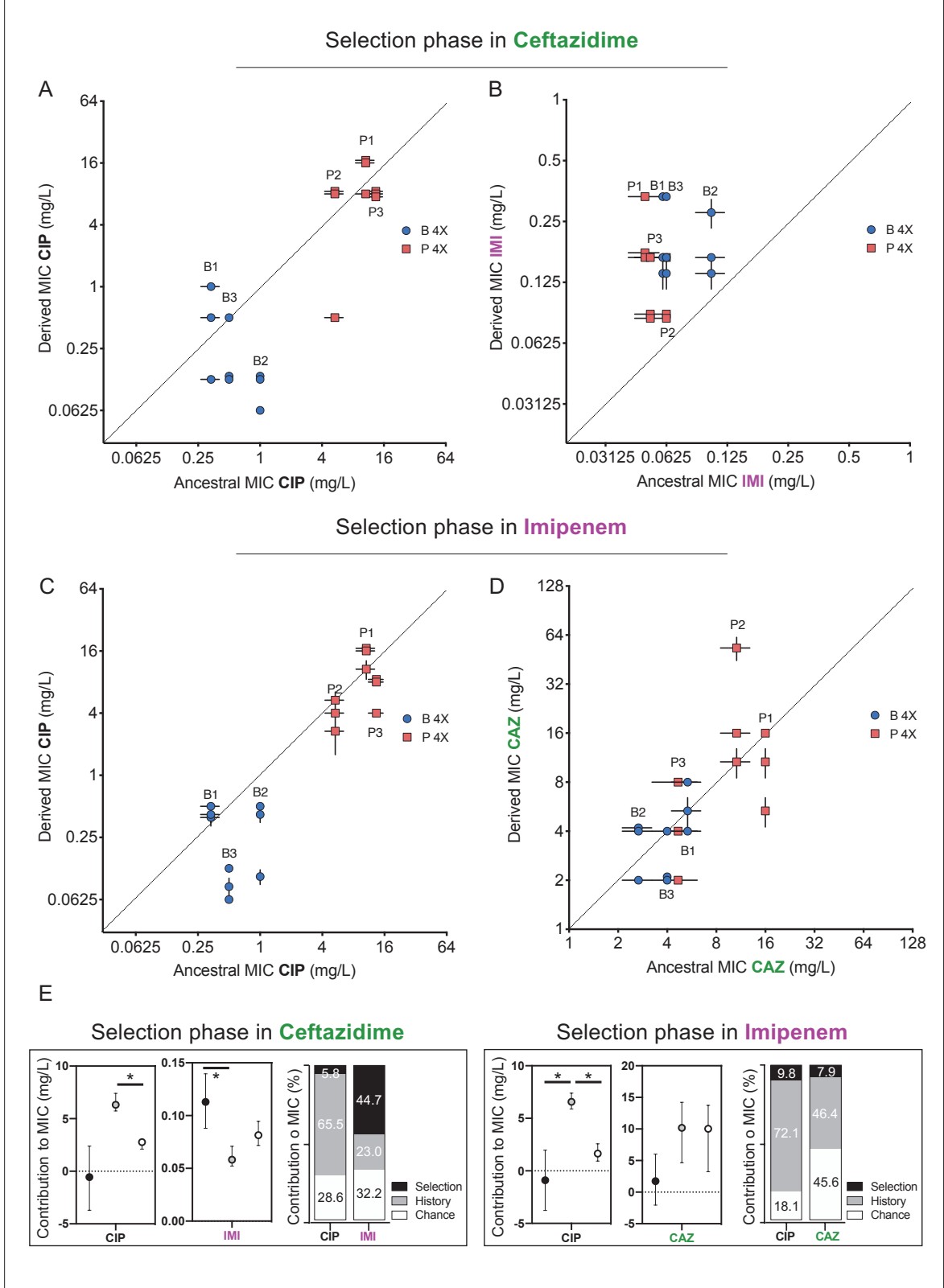

**Figure 3.** Collateral resistance caused by history, chance, and selection. Panel (**A**) shows CIP resistance and (**B**) shows IMI resistance following 12 days of CAZ treatment. Panel (**C**) shows CIP resistance and (**D**) shows CAZ resistance following 12 days of IMI treatment. Blue symbols: populations evolved from B (biofilm-evolved) ancestors; red squares: populations evolved from P ancestors (planktonic-evolved). Some symbols representing identical data points are jittered to be visible. MICs were measured in triplicate and shown ± SEM. (**E**) Contributions of each evolutionary force. Error bars indicate 95 % confidence intervals. Asterisks denote p<0.05.

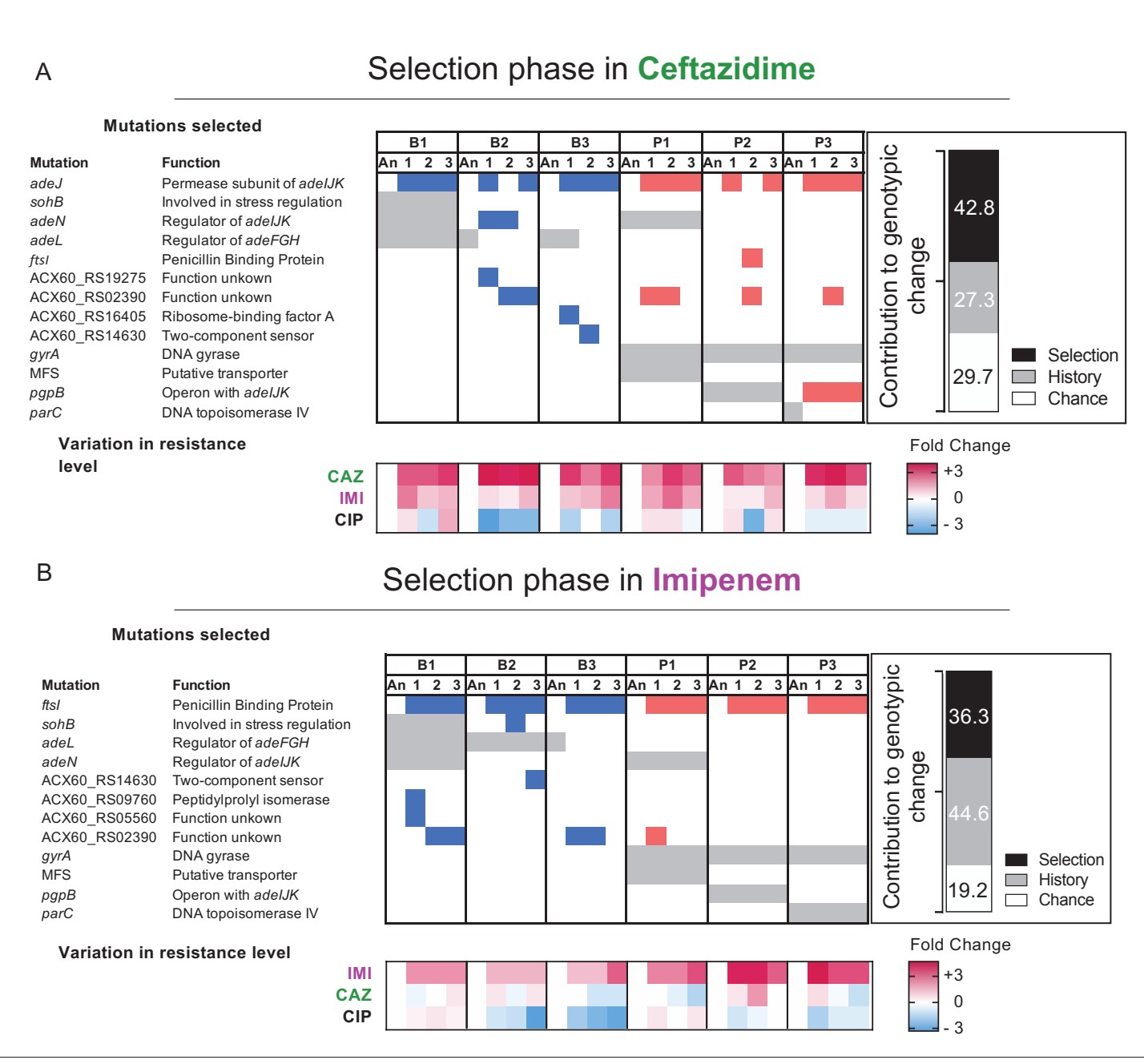

**Figure 4.** Mutated genes in the populations evolving in presence of a new antibiotic. Each column represents a population propagated in CAZ (**A**) or in IMI (**B**). Grey shading indicates the mutated genes present in the ancestral clones derived from the "history phase". Blue and red denote mutated genes after the 'selection phase' in CAZ or IMI and if those lines experienced prior planktonic selection (red) or biofilm growth (blue). Only genes in which mutations reached 75 % or greater frequency or that became mutated in more than one population are shown here. A full report of all mutations is in *Figure 4—source data 1*. The relative contributions of history, chance, and selection to these genetic changes are shown in the insets. Below: log₂ changes in evolved resistance for each population shown as a heatmap summarizing the data from *Figures 2 and 3*.

The online version of this article includes the following figure supplement(s) for figure 4:

**Source data 1.** Putative driver mutations and resistance levels of the replicate populations after 12 days evolving in presence of CAZ or IMI.

**Source data 2.** Complete list of mutated genes from the sequenced populations and clones.

**Figure supplement 1.** Mutated genes in the P3 evolving in the presence of CAZ.

also regulated by *adeN* (*Figure 1D*; *Santos-Lopez et al., 2019*). Subsequently, evolution in increasing concentrations of CAZ selected at least one mutation in *adeJ* in 16/18 populations (*Figure 4A*); this gene encodes the permease subunit of AdeIJK that is a known cause of CAZ resistance (*Lee et al., 2017*). The two exception populations instead acquired mutations in *adeN*, in ACX60_RS2390, a gene of unknown function, and in *ftsI*, the target of CAZ. Evolution in IMI also selected mutations in the *ftsI* gene in all populations (*Figure 4B*); this gene encodes penicillin binding-protein 2, one of the most common causes of de novo resistance to IMI in clinical isolates (*Lee et al., 2017*). Therefore, evolution in β-lactam antibiotics generated convergent evolution regardless of the genetic background (*Vogwill et al., 2014*; *Scribner et al., 2020*).

Yet despite these genetic similarities, replicate populations reached different resistance levels (*Figure 2B,D*). As the resistant phenotype was measured in mixed populations with diverse genetic backgrounds, it is possible that even though a resistance allele is fixed, different genotypes within each population could explain the phenotypic differences. Evidence of this heterogeneity might be seen when comparing the five replicate IMI populations that acquired the same mutation in *ftsI* (A579V) but differ in resistance levels by up to fourfold (*Figure 4* and *Figure 1—source data 1*). Another potential explanation for different phenotypes associated with mutations in the same gene is that different mutations may produce different resistance levels. Evidence for this possible explanation is seen when comparing replicate populations derived from ancestor P1, where different SNPs in *adeJ* (*Figure 4*, *Figure 1—source data 2*) produce varied resistance (*Figure 2*), perhaps by altering the function of this permease in different ways. Follow-up experiments with reconstructed variants in isogenic backgrounds are needed to test this hypothesis. To summarize, both varied pleiotropy of different mutations in the same drug targets and interactions between mutations in different drug targets may constrain AMR evolution.

## Collateral sensitivity resulting from genetic reversions

Antibiotic resistance mutations typically incur a fitness cost that favor sensitive strains in the absence of antibiotics. Phenotypic reversion to sensitive states is commonly caused by secondary mutations in other genes (*Durão et al., 2018*; *Dunai et al., 2019*), but it could also be caused by genotypic reversions in which the ancestral allele is selected under drug-free conditions (*Teotonio and Rose, 2000*; *Bedhomme et al., 2013*; *Rebolleda-Gomez and Travisano, 2019*). In our experimental system, assuming a conservative uniform distribution of mutation rate of $10^{-3}$/genome/generation (*Lynch et al., 2016*), each base pair experiences approximately three mutations on average during the 12 days of serial transfers (*Santos-Lopez et al., 2019*). This estimate implies that reversion mutations affecting historical CIP resistance did occur amidst billions of cell divisions, but nonetheless they are expected be much rarer than suppressor mutations in other genes. Surprisingly, we identified genetic reversion of *adeL* mutations five different times in CAZ lines and three different times in IMI lines (*Figure 4A,B*, respectively), and these back-mutations reversed resistance tradeoffs between β-lactams and CIP (*Figures 3A and 4A* for CAZ, *Figures 3C and 4B* for IMI). We also observed genetic reversion of *parC* mutations in each P3 replicate propagated in CAZ (*Figure 4A*). The topoisomerase IV *parC* is one of the canonical targets of CIP but these mutations have been shown to incur a high fitness cost in the absence of CIP (*Kugelberg et al., 2005*). Selection in the presence of CAZ or IMI therefore favored these reversions in the absence of CIP, but in this case without notable loss of CIP resistance presumably via secondary mutations in *pgpB* (*Figure 4A*, *Santos-Lopez et al., 2019*). It can be argued that we propagated polygenic colonies bearing the resistant genotype and the sensitive genotype at very low frequencies but undetectable by our analysis methods. For example, we detected standing genetic variation in *adeL* in the B2 ancestral clone that could explain the reversion to the sensitive genotype. However, with a depth of ca. 300 × coverage, we did not detect any low frequent variants either in B3 or P3 that could explain the reversions. To test the unlikely possibility that the sensitive allele was present in the ancestral clone, we re-isolated the P3 ancestral clone, selected a single clone, and propagated it again in increasing concentrations of CAZ. By re-plating the ancestral clone , we reduced the possibility that the sensitive allele was present at low frequencies in the new selected clone. At the end of the experiment, we detected the *parC* reversion in one out of three evolved lines (*Figure 4—figure supplement 1*), confirming that the sensitive allele arose by chance and was selected for in presence of CAZ. The high frequency of mutational reversion observed in these experiments indicates that these resistant

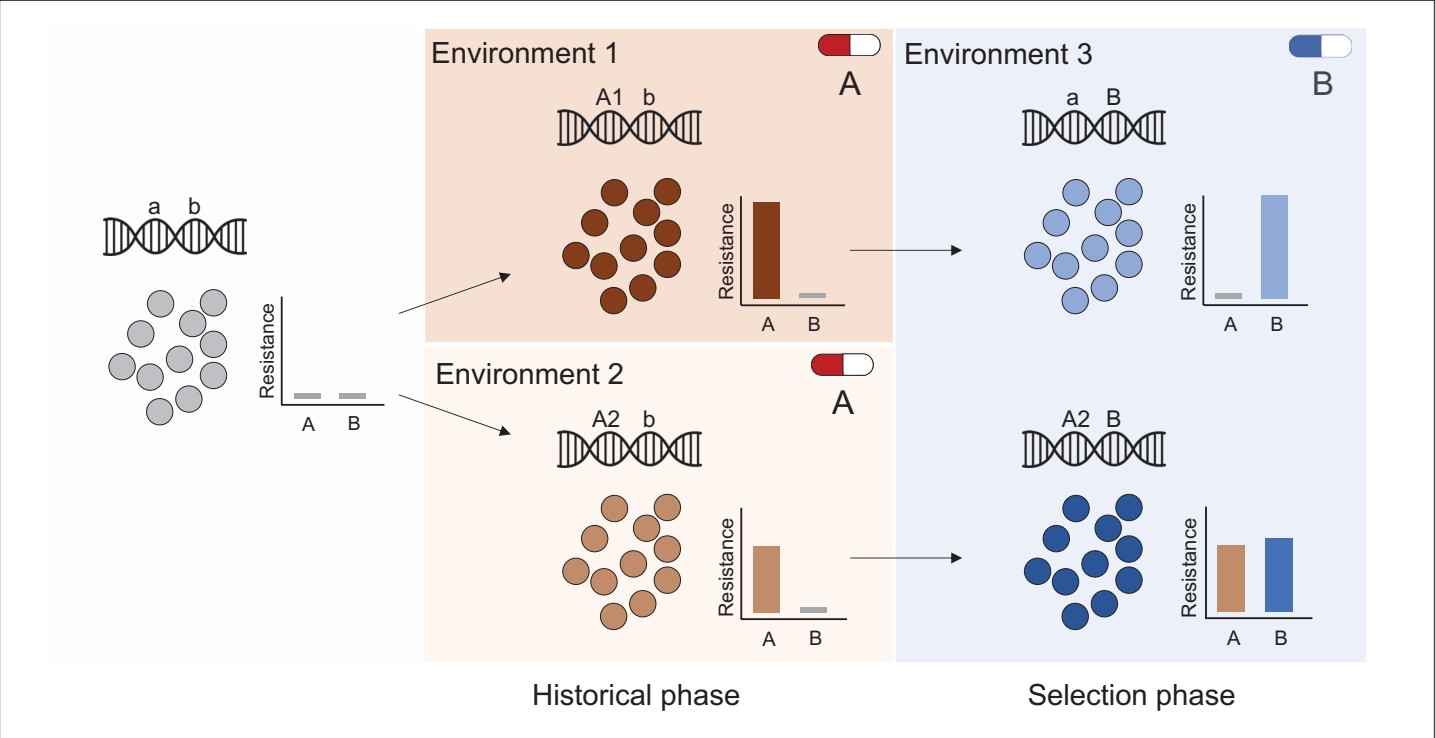

**Figure 5.** Evolutionary history and natural selection determine the evolution of antibiotic resistance. A sensitive population (left panel) is subjected to two successive treatments (antibiotic A and antibiotic B, middle and right panels respectively). First, the population was treated with antibiotic A in either of two different environments (middle panel top and bottom) that selected different genotypes (mutations A1 and A2) with distinct resistance phenotypes (middle panel insets). During subsequent exposure to a second antibiotic (**B**), this evolutionary history determined resistance levels (right panel) to both drugs A and B, for instance resulting in the loss of resistance to drug A (top right panel).

determinants are under enormous constraint and impose fitness costs in the presence of CAZ or IMI (*Pennings et al., 2021*).

## Discussion

Stephen Jay Gould famously argued that replaying the tape of life is impossible because historical contingencies are ubiquitous (*Gould, 1990*). The evolution and spread of AMR provide a test of this hypothesis because countless evolution experiments are initiated each day with each new prescription to combat infections caused by bacteria with different histories. Previous studies suggest that the predictability of antibiotic resistance – or the fidelity of the replay – depends on the pathogen, the antibiotic treatment, and the growth environment (*Vogwill et al., 2014*; *Gifford et al., 2018*; *Wistrand-Yuen et al., 2018*; *Card et al., 2019*; *Santos-Lopez et al., 2019*; *Scribner et al., 2020*). Here, we have quantified contributions of history, chance, and selection to AMR evolution, using six different ancestors replicated in each of two different antibiotic treatments. In the end, selection is unsurprisingly the predominant force in the evolution of AMR and produced convergent evolution even at the nucleotide level in some instances. Yet history and chance play clear and important roles in the emergence of new resistance phenotypes (*Figures 3B,D and 5*, *Vogwill et al., 2014*), the extent of evolved resistance (*Figures 2 and 3*), the generation of collateral sensitivity networks, (*Pal et al., 2015*), and the predictability of the final resistance phenotype (*Figures 1 and 4*, *Gifford et al., 2018*; *Scribner et al., 2020*). If we consider that the established history of these experimental populations is shallow – the result of only 80 prior generations of growth in a different antibiotic that selected between one and three mutations – it is remarkable how deeply these genotypes were imprinted, resulting in divergent evolutionary trajectories under stringent selection in new drugs. Our data also suggest that, as in *Drosophila* (*Teotonio and Rose, 2000*), viruses (*Bedhomme et al., 2013*) and

yeast (*Rebolleda-Gomez and Travisano, 2019*), history and chance may determine the reversibility of acquired traits (*Figure 5*).

This probability of reversion is potentially clinically important because exploitable collateral sensitivity networks can arise, such as the tradeoff between CIP resistance and β-lactam resistance identified here (*Pal et al., 2015*). Finally, our data reveals that evolution of AMR follows a clear diminishing return pattern, where antibiotic pressure selects for mutations with progressively smaller phenotypic effects as the population is treated with higher antibiotic concentrations (*Figure 2—figure supplement 1*). This result mirrors findings in the original Travisano et al. paper (*Travisano et al., 1995*), where populations that were pre-adapted to compete well in maltose did not adapt further, but populations with major deficiencies in maltose evolved to become just as fit. This result may be instructive for AMR management: on the one hand, more resistant populations at the outset did not increase this phenotype further, but on the other hand, more susceptible lines rapidly compensated for this deficit.

Our experiment was performed in planktonic cultures and was limited to a sensitive strain of *A. baumannii* treated with a single fluoroquinolone followed by one of two β-lactam drugs. These were deliberate experimental design choices that allowed careful assessment of the evolutionary forces at play in a rapidly evolving population but may be considered limitations for some broader applications. Despite these limitations, our finding that history and chance are ancillary forces compared to the strength of selection imposed by antibiotics is universal and is well supported by the literature. For instance, exposure to fluroquinolones in Gram-positive or Gram-negative bacteria commonly selects for mutations in *gyrA* (*Seward and Towner, 1998*; *Weigel et al., 1998*; *Hooper and Jacoby, 2015*). However, we also observed that history and chance can play important roles in resistance evolution in certain specific environments. For example, the reversions in *adeL* are probably lifestyle dependent and would not be expected to occur if we replay the experiment in the biofilm lifestyle instead of planktonic.

Finally, our experiment focuses solely on de novo mutations and does not allow the opportunity for horizontal gene transfer from other species or strains, which is the principal mechanism of the emergence of AMRs in most clinical settings (*MacLean and San Millan, 2019*). However, genetic background also affects the fitness of transmissible elements (*Alonso-del Valle et al., 2021*) and epidemiological data indicate that evolutionary history constrains the persistence of resistance mediated by plasmids (*Dunn et al., 2019*; *León-Sampedro et al., 2021*). The framework defined here illustrates the potential to identify genetic and environmental conditions where selection is the most dominant evolutionary force and it predictably produces antagonism between resistance traits. With ever greater knowledge of the present state, we gain hope for guiding the future to exploit the past.

# Materials and methods
## Summary of experimental design
Following *Travisano et al., 1995*, consider replicate populations founded by a single clone that are propagated in the same environment for a certain number of generations. We can dissect the roles of each evolutionary force by measuring changes in the mean and variance of an important trait (e.g., fitness or antibiotic resistance) (*Figure 1A*). In the first scenario, the mean and variance of the studied trait did not change, so one can conclude that the trait did not evolve (Top left panel, *Figure 1A*). In the second scenario, while the grand mean of the trait remains the same as the ancestral value, trait variance increases (top middle panel, *Figure 1A*). Here, the main evolutionary force is chance, comprised of mutation and genetic drift. In the third scenario, the grand trait mean increases significantly, but not the variance (top right panel, *Figure 1A*), a change that is best explained by natural selection. Combining these two forces of chance and natural selection, we would expect both trait mean and variance to increase (bottom left panel, *Figure 1A*). Note that these four scenarios describe outcomes when starting from a single clone, that is with no genetic variation, but this rarely happens in nature. If we conduct the same experiment using different ancestors that vary in the studied trait, two additional scenarios are possible. In the first, the initial variation among the different ancestors is erased by chance and adaptation (bottom middle panel, *Figure 1A*), which cause the trait variance and mean to increase to identical values, regardless of the ancestral value. In the last scenario, the effect of history constrains the evolution of the trait, where the final trait value correlates with the

ancestral value (bottom right panel, *Figure 1A*) despite contributions of both chance (increased variance) and selection increasing the trait.

## Experimental evolution

### Historical phase

Before the start of the antibiotic evolution experiment, we planktonically propagated one clone of the susceptible *A. baumannii* strain ATCC 17978-mf (*Figure 1—figure supplement 1*) in a modified M9 medium (referred to as M9$^+$) containing 0.37 mM CaCl$_2$, 8.7 mM MgSO$_4$, 42.2 mM Na$_2$HPO$_4$, 22 mM KH2PO$_4$, 21.7 mM NaCl, 18.7 mM NH$_4$Cl, and 0.2 g/L glucose and supplemented with 20 mL/L MEM essential amino acids (Gibco 11130051), 10 mL/L MEM nonessential amino acids (Gibco 11140050), and 10 mL each of trace mineral solutions A, B, and C (Corning 25021–3 Cl). This preadaptation phase was conducted in the absence of antibiotics for 10 days (ca. 66 generations) with a dilution factor of 100 per day. All experimental evolutions described here – preadaptation, historical phase and selection phase – were performed in 18 mm glass tubes containing 5 mL of M9$^+$.

After 10 days of preadaptation to M9$^+$ medium, we selected a single clone and propagated for 24 hr in M9$^+$ in the absence of antibiotic. We then subcultured this population into 20 replicate populations. Ten of the populations (5 planktonic and 5 biofilm) were propagated every 24 hr in constant subinhibitory concentrations of CIP, 0.0625 mg/L, which corresponds to 0.5 × the minimum inhibitory concentration (MIC). We doubled the CIP concentrations every 72 hr until 4 × MIC (*Figure 1B*).

### Selection phase

Upon the conclusion of the 'historical phase', we selected one clone from three populations previously adapted in biofilm and three populations previously adapted in planktonic conditions. We streaked the populations on ½ Tryptic soy agar (Difco Laboratories Inc, NJ) and selected one clone per population that were sequenced as explained later, growing during 24 hr in M9$^+$. Clone B2 was found to contain standing genetic variation after 24 hr growing in M9$^+$ (*Figure 4—source data 1*). We determined their resistance level to CIP, CAZ, and IMI. Then, we propagated planktonically each clone independently with a dilution factor of 100 or in the presence of increasing concentrations of CAZ or in increasing concentrations of IMI. For each population, we used their own MIC to CAZ or IMI to determine the concentrations used in this phase (*Figure 1—source data 1*). We serially passaged 50 µL into 5 mL of M9$^+$ which corresponds to approximately 6.64 generations per day. The average population size at day 1 was $4.7 \times 10^8$ ($\pm 1.1 \times 10^8$) CFU/mL and $2.8 \times 10^9$ ($\pm 1.4 \times 10^9$) at day 12. As a control, we propagated two replicates of the pre-adapted *A. baumannii* clone in the absence of antibiotics for 12 days. We froze 1 mL of the propagated populations at days 1, 3, 4, 6, 7, 9, 10, and 12 in 9 % of DMSO.

## Antimicrobial susceptibility characterization

We determined the MIC of CAZ, CIP, and IMI of the whole population by broth microdilution in Mueller-Hinton as explained before according to the Clinical and Laboratory Standards Institute guidelines (*Santos-Lopez et al., 2019*), in which each bacterial sample was tested in twofold-increasing concentrations of each antibiotic. To perform the MICs, we streaked the ancestral clones and the evolved populations in ½ Tryptic soy agar (Difco Laboratories Inc, NJ) without antibiotics. For clones, we selected three to five clones and resuspended them in PBS, and for the populations, we took a full loop of the frozen biomass to obtain a representation of the whole population. In order to follow the CLSI standards, both the clones and the populations were diluted to a 0.5 MacFarland units. Then, we diluted the PBS containing bacteria 1/10 times in Mueller–Hinton broth and performed the MICs as recommended by the CLSI guidelines. The CIP, CAZ, and IMI were provided by Alfa Aesar (Alfa Aesar, Wardhill, MA), Acros Organics (Across Organics, Pittsburgh, PA), and Sigma (Sigma-Aldrich Inc, St. Louis, MO), respectively.

## Genome sequencing

We sequenced the two replicate drug-free passaged controls, six ancestral clones, and whole populations of the 36 evolving populations (18 evolved in the presence of CAZ and 18 evolved in the presence of IMI) at the end of the experiment. We revived each population or clone from a freezer stock in the growth conditions under which they were isolated (i.e. 5 mL of M9$^+$ in 18 mm glass tubes adding

the same CAZ or IMI concentration which they were exposed to during the experiment) and grew for 24 hr. We centrifuged 1 mL of the ON culture, and we extracted DNA using the Qiagen DNAeasy Blood and Tissue kit (Qiagen, Hiden, Germany) following the indications from the manufacturers. The sequencing library was prepared as described by *Turner et al., 2018* according to the protocol of *Baym et al., 2015*, using the Illumina Nextera kit (Illumina Inc, San Diego, CA) and sequenced using an Illumina NextSeq500 at the Microbial Genome Sequencing Center. The mutations detected in the drug-free passage controls (*Figure 4—source data 2*) were subtracted from subsequent analyses.

## Statistical analysis of the role of each evolutionary force

We calculated the phenotypic effect of the evolutionary forces using a nested linear mixed model. By means of this nested linear mixed model including ancestors and replicates as random effects, we estimated the effect of history as the square root of the variance among all propagated populations; the effect of chance as the square root of the variance between the replicates propagated from the same ancestor; and the effect of selection was calculated as the difference in grand mean of the propagated replicates and their ancestors (*Figure 2—source data 1*).

Percentile bootstrap was employed to compute the confidence intervals of each force at the level of significance $\alpha = 0.05$ by taking 1000 random samples with replacement. In addition, the statistical evidence of each force was assessed adopting a Bayesian approach, which allows to circumvent the issues associated to null hypothesis statistical testing (*Wagenmakers, 2007*). Specifically, a set of models excluding each force (Null hypotheses) were confronted against the full model including the three forces (Alternative Hypothesis). Thus, let $BIC_1$ be the Bayesian Information Criterion associated to the alternative model and $BIC_0$ the Bayesian Information Criterion for one of the null models. Then, a Bayes factor can be approximated as follows:

$$BF_{10} \approx \frac{Pr(D|H_1)}{Pr(D|H_0)} = exp\left(\left(BIC_0 - BIC_1\right)/2\right)$$

where $Pr(D|H_0)$ and $Pr(D|H_1)$ are the marginal probabilities of the data under the null and alternative models respectively. Hence, the Bayes factor allows to quantify how likely the inclusion of a force is with respect to its absence according to the observed data. All these estimations were performed using *blme* v1.0–4 R package (https://cran.r-project.org/package=blme). All values were normalized to one to calculate the influence of each evolutionary force.

The roles of the evolutionary forces at the genotypic level were calculated using all identified mutations above a detection threshold of 5 % based on the Manhattan distance ($d_M$) between populations. For a pair of populations *j* and *k* with *n* genes,

$$d_M = \sum_{i=1}^{n} \left|x_{ij} - x_{ik}\right|$$

where $x_{ij}$ is the frequency of mutated alleles in gene i in population j, relative to the *A. baumannii* strain ATCC 17978-mff. For a given gene, $x_{ij} - x_{ik}$ is zero if there are no mutations present in that gene in either population j or k or if the frequency of mutated alleles is the same in both populations. If multiple mutations in a given gene were present in a population, the frequency of mutated alleles was the sum of the frequencies of all mutated alleles in that gene. This assumes that each mutation occurred on a different genetic background.

The genotypic role of chance was calculated as half the mean $d_M$ between all pairs of evolved populations founded from the same ancestral clone. The genotypic role of history was calculated as half the mean $d_M$ between all pairs of evolved populations founded from the different ancestral clones minus the role of chance. The genotypic role of selection was calculated as the mean $d_M$ between evolved populations and their founding clone, minus the roles of chance and history. In comparing the role of the different forces, we accounted for the fact that chance and history are calculated as the distance between two evolved populations, whereas selection is calculated as the distance between ancestral and evolved populations, by defining the roles of chance and history as half the mean $d_M$. In calculating selection, mutations present in the founding clone were not excluded when subtracting the effect of history.

To analyze the role of each force, it is important to note some limitations of the study. First, the analysis of the forces makes no assumption about the linearity or additivity of their effects. Phenotypic variation between populations is simply partitioned between three possible pools: differences

between ancestral and evolved populations (selection), differences between evolved populations with different ancestors (history), and differences between evolved populations with the same ancestor (chance). For the genotypic metric, the same logic applies and differences in the frequencies of mutations are attributed to the same three pools. Where non-additive effects become important to consider is in interpreting the differences between the phenotypic and genotypic metrics. The contributions of the forces at the genotypic and phenotypic levels would be the same if every mutation that arose had an equal effect on the phenotype (or at least that the frequency of each mutation in the population was proportional to its phenotype) and phenotypic effects were additive, with no epistasis. The greater the deviation from those assumptions, the greater the differences will be between the genotypic and phenotypic roles of history, chance, and selection. Second, while the three forces play ongoing roles during evolution, it is important to note that the moment when we analyze their role has been arbitrarily selected. For instance, historical effects are cumulative and every moment in the course of evolution may be contingent on previous historical adaptations (*Travisano et al., 1995*). Here, we analyze how evolution in two lifestyles, planktonic and biofilm, challenged by one antibiotic, CIP, influences further adaptation to a second antibiotic, CAZ or IMI. Therefore, we consider evolutionary history to any adaptation occurred before exposure to CAZ or IMI, and we measured the role of the forces at only two timepoints: after 3 or 12 days exposing the populations to the antibiotic.

All statistical comparisons of MIC values were performed on the $\log_2$ transformed values. Differences in grand means between populations were analyzed by a one-way nested ANOVA with Tukey's multiple comparison tests or by a nested t-test. Spearman correlation was performed using the grand means to determine the correlation between the ancestral MIC and the fold change of MIC acquired during the experiment. There are three possible outcomes by correlating the original MIC and the fold dilution change: (1) a negative correlation, in which the populations with lower initial MICs increased their resistance level more than populations with higher MICs, implies that the selection erased the previous effects of history; (2) a positive correlation indicates that initial differences in MIC were magnified by selection; and (3) a lack of correlation indicates that the effect of history did not change before and after selection.

## Data processing

The variants were called using the breseq software v0.31.0 (*Barrick et al., 2014*) using the default parameters and the -p flag when required for identifying polymorphisms in populations after all sequences were first quality filtered and trimmed with the Trimmomatic software v0.36 (*Bolger et al., 2014*) using the criteria: LEADING:20 TRAILING:20 SLIDINGWINDOW:4:20 MINLEN:70. The version of *A. baumannii* ATCC 17978-mff (GCF_001077675.1 downloaded from the NCBI RefSeq database,17-Mar-2017) was used as the reference genome for variant calling. We added the two additional plasmid sequences present in the *A. baumannii* strain (NC009083, NC_009084) to the chromosome NZ_CP012004 and plasmid NZ_CP012005. Mutations were then manually curated and filtered to remove false positives under the following criteria: mutations were filtered if the gene was found to contain a mutation when the ancestor sequence was compared to the reference genome or if a mutation never reached a cumulative frequency of 10 % across all replicate populations.

## Acknowledgements

We thank Tim Cooper, Alvaro San Millan, Roderich Röhmild, and Sergio Santos for helpful discussions and proofreading of the paper, Dan Snyder for laboratory assistance and Christopher Deitrick for depositing the sequences in the NCBI database. Funding: This work was supported by the Institute of Allergy and Infectious Diseases at the National Institutes of Health (grant U01AI124302 to VSC) and by the European Comission (H2020-MSCA-IF-2019, 895671-REPLAY to AS-L).

## Additional information

### Funding

| Funder | Grant reference number | Author |
|---|---|---|
| National Institutes of Health | U01AI124302 | Vaughn S Cooper |
| Horizon 2020 | H2020-MSCA-IF-2019 REPLAY-895671 | Alfonso Santos-Lopez |

The funders had no role in study design, data collection and interpretation, or the decision to submit the work for publication.

### Author contributions

Alfonso Santos-Lopez, Conceptualization, Data curation, Formal analysis, Investigation, Methodology, Validation, Visualization, Writing – original draft, Writing – review and editing; Christopher W Marshall, Formal analysis, Methodology, Software, Visualization, Writing – original draft, Writing – review and editing; Allison L Haas, Formal analysis, Investigation, Methodology; Caroline Turner, Formal analysis, Methodology; Javier Rasero, Formal analysis, Methodology, Software; Vaughn S Cooper, Conceptualization, Funding acquisition, Project administration, Supervision, Writing – original draft, Writing – review and editing

### Author ORCIDs

Alfonso Santos-Lopez (iD) http://orcid.org/0000-0002-9163-9947
Christopher W Marshall (iD) http://orcid.org/0000-0001-6669-3231
Allison L Haas (iD) http://orcid.org/0000-0002-2154-4328
Caroline Turner (iD) http://orcid.org/0000-0003-1347-518X
Javier Rasero (iD) http://orcid.org/0000-0003-1661-2856
Vaughn S Cooper (iD) http://orcid.org/0000-0001-7726-0765

### Decision letter and Author response

Decision letter https://doi.org/10.7554/eLife.70676.sa1
Author response https://doi.org/10.7554/eLife.70676.sa2

## Additional files

### Supplementary files

• Supplementary file 1. List of deposited sequences from clones and populations and the corresponding accession numbers.

• Transparent reporting form

### Data availability

All data generated or analyzed in this study are included in the manuscript, supporting files, or at https://github.com/sirmicrobe/chance_history_selection, where raw experimental values and statistical analysis code is shared. All sequences were deposited into NCBI under the BioProject number PRJNA485123 and accession numbers can be found in Supplementary file 1.

The following dataset was generated:

| Author(s) | Year | Dataset title | Dataset URL | Database and Identifier |
|---|---|---|---|---|
| Santos-Lopez A, Marshall CW | 2020 | Predicting the emergence of antibiotic resistance | https://www.ncbi.nlm.nih.gov/sra?linkname=bioproject_sra_all&from_uid=485123 | NCBI Sequence Read Archive, PRJNA485123 |

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
