## [Decision Letter]

**Acceptance summary:**

This work uses an elegant and well-designed experimental evolution strategy to investigate the roles of history, chance, and selection on the evolution of antibiotic resistance in the clinical pathogen *Acinetobacter baumannii*. The authors show that while history impacts the evolution of antimicrobial resistance, this effect decreases with increasing drug selection strength and indicates that natural selection is a dominant driver. The work presents clear, unambiguous data on the importance of antibiotic exposure on the evolution of resistance.

**Decision letter after peer review:**

Thank you for submitting your article "The roles of history, chance, and natural selection in the evolution of antibiotic resistance" for consideration by *eLife*. Your article has been reviewed by 3 peer reviewers, and the evaluation has been overseen by a Reviewing Editor and Dominique Soldati-Favre as the Senior Editor. The following individual involved in review of your submission has agreed to reveal their identity: Alan McNally (Reviewer #2).

Essential revisions:

While the work addresses a very relevant and interesting question, the reviewers raised some concerns that need to be addressed in order to improve the manuscript. Additional experimental details should be included, and clarification is needed regarding some of the concepts used. More importantly, the limitations of the approach need to be specifically addressed and some of the claims and interpretations should be toned down, particularly given the absence of a drug-free control.

1) A main concern is in the interpretation of populations having the same genetics but different phenotypes (eg lines 300-310) – this conclusion is wrong given the experimental design. The authors provide several explanations but leave out a high likelihood alternative which is that different experiments consist of mixed populations with diverse genetic makeups. Alternatively, if the authors meant instead to focus on specific mutations (ie same mutations/localized mutations) regardless of the background, then this conclusion is trivial – it is known that pleiotropy exists.

2) I agree with the author's definitions of change due to history vs change due to selection given the evolution they designed – and I found this quite innovative and overall strong. However, the definition and calculations of evolution due to chance here are shaky given the three replicates only. There are plenty of extremely rigorous estimates of genetic drift during bacterial evolution (see all of Lenski's work) – three populations is quite small to make that claim and run the statistics. Including chance as a possible contributor didn't add much to the story, and it raised more questions than answers given the extremely diverse genetics that were observed, and that data of the mixed populations below 75% was not included.

3) Experiments are flawed without a no drug control. Was the reversion of CIP resistance specific to the new drug selection, or is it simply the lack of CIP pressure? This is important as the authors make claims about drug-specific evolutionary tradeoffs and collateral sensitivity. The methods in line 428: "We froze 1 mL of the control populations on days…" but this is the only place a control population is mentioned.

4) Much of the data was difficult to follow, such as the fold changes for MICs – for example, following the discussion of CIP resistance, maintenance, and reversion was nearly impossible with the reference to figure 4 fold changes only. Also going back and forth between IMI and CAZ, which is discussed in parallel near the beginning, but then broken down later on, is another example.

5) Key experimental details are missing. What volume were the populations propagated in? Population experiments were not well described – how were replicate MICs initiated? How much volume was used for sequencing? So on and so forth. These are particularly important to interpret the results.

6) Why was a 75% threshold used to determine alleles if the populations were sequenced to a level of 5%?

7) Despite being described in great detail in the methods, it doesn't come across to non-modeler such as myself exactly how selection, history, and chance were quantified in the phenotypic experiments shown in figure 2. I am a bit clearer as to how these were quantified for the genomic data. However, I think a broad readership may benefit from a layman's description of how these were differentiated. It is vital to the science and data shown and I would certainly have appreciated a clarification, especially for the phenotypic data in figure 2.

8) In line 307. In the absence of bacterial genetic experiments to confirm that these historical infections are actually driving the 4 fold differences in phenotype, I think this inference needs to be toned down.

9) Line 343: Is there a benefit to be had of fitness experiments in antibiotic-free medium to confirm the supposition made here?

10) In some cases, I found the framing of the results in terms of history, chance, and selection to be a bit overly general, which sometimes obscured the specific results being reported. The paper could be improved by using more specific language-perhaps restricted in scope- in describing and interpreting the results, both because 1) it's not obvious to me that the results would apply generally to antibiotic resistance beyond the very nice, but potentially system-specific, results presented here; and 2) the terms themselves (history, chance, selection) could conceivably have different meanings in different contexts (more on this below).

11) The study design has been used in numerous previous studies; it is well established, elegant, and has given rise to many new insights. However, as I understand it, there are some inherent assumptions of the approach that should be briefly discussed. Most notably, does the approach inherently assume that the effects of history, chance, and selection are additive (or perhaps linear) in some sense (in terms of phenotypic variance measurements or the Manhattan-based genotype metric)? While this simplifying assumption seems critical to the power of the approach, it is not clear to me that this assumption holds in general. When I try to think of this in terms of, say, a simple population dynamics model, the terms history, chance and selection are themselves somewhat nebulous, and it's not clear to me that they could be unambiguously and uniquely defined even in simplified theoretical models (or more directly, that the variance-based phenotype measures correspond to well-defined features or parameters of population dynamics models). I say this not to criticize the approach-again, its power lies in the simplicity of the design and the intuitive value of separating these three evolutionary features and attempting to quantify their contributions. But I think the article could be strengthened by briefly discussing the underlying assumptions-ideally by pointing to previous work (if it exists) that establishes that the features are additive and measurable in the sense required by the experimental design. If not, I think it would be worth discussing that limitation briefly, as I worry the inherently nonlinear nature of these very complicated, evolving systems could lead us to misinterpret the results. Given the general success of similar approaches in past work, I suspect the authors have thought through these issues in detail; discussing those points might open the paper to a broader audience not intimately familiar with all the previous studies.

12) One example related (but not identical to) the point above: in the current experiment, the role of history is defined in terms of previous selection conditions (drug and growth phase). But the new evolution experiment itself has multiple time points, and even qualitatively distinct epochs (sub- and super-MIC drug). So one might argue that history is playing a role continuously throughout the experiment-history not merely of the previous selection in fluoroquinolones, but also history of the previous time points / epochs of the new evolution (β-lactams). My point is that it is important to clearly define the terms at all stages and discuss, at least briefly, the limitations of the definitions that are chosen.

13) While history, chance, and selection are quantified at both the genetic and phenotype level, it's not clear to me that these numbers can be directly compared to one another (though it's tempting to do so!). Could the authors briefly comment on the connection between these measures-that is, when (and to what extent) one would expect correlations between them (e.g. high levels of historical influence at the genetic level leads to high levels of historical influence at the phenotype (MIC) level….assuming the definitions used here).

14) Do the authors have any thoughts on how the results might be affected by the fact that the new evolution experiments take place in planktonic (rather than biofilm) conditions? How might the results differ if they had been performed in biofilm, and what could you learn from the fully symmetric experiment (P/ B initial strains evolved in both P and B new selection conditions). The authors may wish to discuss this avenue for future work.

---

## [Author Response]

Essential revisions:While the work addresses a very relevant and interesting question, the reviewers raised some concerns that need to be addressed in order to improve the manuscript. Additional experimental details should be included, and clarification is needed regarding some of the concepts used. More importantly, the limitations of the approach need to be specifically addressed and some of the claims and interpretations should be toned down, particularly given the absence of a drug-free control.

We thank the reviewers and editors for their careful consideration of our manuscript. We agree with all reviewer comments and edited the paper accordingly. We have revised the manuscript for general clarity, and we believe that the revised manuscript has been improved through the changes proposed by the reviewers. We have included the missing experimental details, we have softened some conclusions, and we have included two sections about our assumptions and limitations of the study. We also clarified that we propagated drug-free controls, which was not clear in the previous manuscript. We thank the editor and reviewers for improving this manuscript.

1) A main concern is in the interpretation of populations having the same genetics but different phenotypes (eg lines 300-310) – this conclusion is wrong given the experimental design. The authors provide several explanations but leave out a high likelihood alternative which is that different experiments consist of mixed populations with diverse genetic makeups. Alternatively, if the authors meant instead to focus on specific mutations (ie same mutations/localized mutations) regardless of the background, then this conclusion is trivial – it is known that pleiotropy exists

We agree with the reviewer (see also comment #8). We have modified the text acknowledging that we are measuring the phenotype of mixed populations that could be genetically heterogeneous, and we have also toned down the conclusion of this section. (Lines 324-332)

Text explicitly addressing this comment:

"As the resistant phenotype was measured in mixed populations with diverse genetic backgrounds, it is possible that even though a resistance allele is fixed, different genotypes within each population could explain the phenotypic differences" and regarding the effects of specific mutations: "Follow-up experiments with reconstructed variants in isogenic backgrounds are needed to confirm this hypothesis."

2) I agree with the author's definitions of change due to history vs change due to selection given the evolution they designed – and I found this quite innovative and overall strong. However, the definition and calculations of evolution due to chance here are shaky given the three replicates only. There are plenty of extremely rigorous estimates of genetic drift during bacterial evolution (see all of Lenski's work) – three populations is quite small to make that claim and run the statistics. Including chance as a possible contributor didn't add much to the story, and it raised more questions than answers given the extremely diverse genetics that were observed, and that data of the mixed populations below 75% was not included.

We appreciate these important points and are glad to provide more nuance to the argument that the role of chance can be determined with our experimental design.

We agree that three populations seems like a small number of replicates to be able to study the effect of chance. However, we are not analyzing the effect of chance in just three replicates, but rather in 3 replicates of 6 ancestors in two independent experiments. In fact, both the genotype and the phenotype results show that most of the replicates started from a single ancestor followed a unique evolutionary trajectory, highlighting the importance of chance in our experiment. Chance is one of the principal forces determining variance in evolution. In the absence of chance, evolution from a single genotype will lead to the fixation of the most fit genotype in every experiment. We only detect 3 events of parallel evolution at the nucleotide level in replicates coming from the same ancestor (B3 and P2 in CAZ, and P1 in IMI) but each of those populations acquired unique mutations during the experiment. The fact that we don’t detect the same evolutionary trajectories in the replicates from a given ancestor -- a result that would be expected in the absence of chance -- shows that the experimental setup is able to detect the effects of chance in the evolution of AMR.

Once we acknowledge that the role of chance can be detected in our experiment and knowing that the number of replicates is limited, we decided to assess the statistical evidence of each force by means of a Bayesian approach, which works particularly well when sample sizes are small and exhibits some advantages over frequentist methods (Wagenmakers, 2007). Specifically, for a given force we computed the Bayes factor that results from comparing the regression model excluding that force against the full model that includes all three forces. This approach enables assessment of how powerful the included force is with respect to its absence according to the observed data. In the case of chance, the support for its inclusion was very strong in most scenarios. On the other hand, an alternative, traditional analysis using bootstrap showed that the contribution of chance was statistically different to the contribution of history in most of the cases analyzed (Figures 2E and 3E).

Finally, we would like to clarify that we have included in our analysis the data of all mutations detected above 5% of frequency. Please also see point #6.

3) Experiments are flawed without a no drug control. Was the reversion of CIP resistance specific to the new drug selection, or is it simply the lack of CIP pressure? This is important as the authors make claims about drug-specific evolutionary tradeoffs and collateral sensitivity. The methods in line 428: "We froze 1 mL of the control populations on days…" but this is the only place a control population is mentioned.

We understand the concerns of the reviewer and we agree with them. With our experimental setup, we can conclude only that the reversions were produced in planktonic conditions, in the presence of CAZ or IMI and in the absence of CIP. We have clarified this in the text (Line 370).

We understand that the reversions were produced when the populations were exposed to CAZ or IMI, and whether they also happen in the absence of the antibiotic is beyond the scope of the paper. See Dunai et al., eLife 2019, a brilliant example where genotypic reversions in antibiotic-free environments were analyzed.

We did not find it necessary to propagate all six clones in triplicates in the absence of CIP to analyze the role of the evolutionary forces during evolution in a new antibiotic. However, we valued the need for a drug-free control of the overall experiment. Therefore, we propagated control populations of the original ancestor in the absence of antibiotic for 12 days. Two replicates were sequenced and the two intergenic mutations observed in the control populations were subtracted from the propagated lines in the presence of CAZ or IMI. We apologize for this confusion and we have clarified it (Lines 484, 501).

4) Much of the data was difficult to follow, such as the fold changes for MICs – for example, following the discussion of CIP resistance, maintenance, and reversion was nearly impossible with the reference to figure 4 fold changes only. Also going back and forth between IMI and CAZ, which is discussed in parallel near the beginning, but then broken down later on, is another example.

The figure of the fold changes is just a summary of figures 2 and 3 in a different representation. We apologize for the confusion, and to improve the clarity of the manuscript, we have referenced the text and figure 4 (Lines 322, 365, 367 and 372). In addition, to facilitate the reading of the paper, we have outlined which sections describe results from the CAZ evolution and which section describes IMI results (Lines 173-176). We believe that describing both experiments together was more challenging to follow and decided to split them in two parts.

5) Key experimental details are missing. What volume were the populations propagated in? Population experiments were not well described – how were replicate MICs initiated? How much volume was used for sequencing? So on and so forth. These are particularly important to interpret the results.

Thank you. We have carefully reviewed the methods and clarified all missing details (Lines 472-475, 491-494, 503-509, 516-519, 547-548).

6) Why was a 75% threshold used to determine alleles if the populations were sequenced to a level of 5%?

We believe that this confusion comes from the legend of the Figure 4: “Only genes in which mutations reached 75% or greater frequency or that became mutated in more than one population are shown here”. We only used a 75% threshold to represent the mutations in Figure 4. We decided to show just mutations with >75% of frequency for clarity in the figure. All mutations detected in the experiment can be found in Table S3. Importantly, for all the analyses, including the estimations of the forces at the genotypic level, all observed mutations were used (Lines 264-265). We have clarified this in the methods section (Lines 547-548).

7) Despite being described in great detail in the methods, it doesn't come across to non-modeler such as myself exactly how selection, history, and chance were quantified in the phenotypic experiments shown in figure 2. I am a bit clearer as to how these were quantified for the genomic data. However, I think a broad readership may benefit from a layman's description of how these were differentiated. It is vital to the science and data shown and I would certainly have appreciated a clarification, especially for the phenotypic data in figure 2.

We added a brief description of how the forces were differentiated in the main text just before the initial results and in the methods (Lines 168-176).

8) In line 307. In the absence of bacterial genetic experiments to confirm that these historical infections are actually driving the 4 fold differences in phenotype, I think this inference needs to be toned down.

We agree that we cannot definitively determine whether previous mutations (a historical effect) or new mutations acquired during in different genes (chance + selection) are driving the 4-fold changes. Therefore, we have toned down the conclusion (Lines 325 – 336). We have acknowledged this comment with the line:

" Follow-up experiments with reconstructed variants in isogenic backgrounds are needed to confirm this hypothesis"

9) Line 343: Is there a benefit to be had of fitness experiments in antibiotic-free medium to confirm the supposition made here?

This is a really interesting point. We do not believe that fitness experiments in antibiotic-free medium confirm the supposition made in the line 343. Evolution experiments can be viewed as optimum competition experiments. Arguably, these designs are the best experiments to test the fitness of a given genotype, as the genotype of interest is competing not only against one strain like in pairwise competitions, but with all other contending mutations that arise each generation. The high level of parallelism found is the best indicator that the reversions are adaptive in those propagated populations.

We do not know if the reversions produce any benefit in the absence of antibiotics, and whether the reversions are beneficial in the absence of CAZ or IMI is trivial for this experimental setup, as those genotypes were selected in the presence of CAZ or IMI. However, we can confirm that those reversions were adaptive in planktonic conditions, in the absence of CIP, and in the presence of CAZ/IMI.

We have clarified that the reversed mutations initially imposed a fitness cost in presence of CAZ or IMI and in the absence of CIP (Line 388).

10) In some cases, I found the framing of the results in terms of history, chance, and selection to be a bit overly general, which sometimes obscured the specific results being reported. The paper could be improved by using more specific language-perhaps restricted in scope- in describing and interpreting the results, both because 1) it's not obvious to me that the results would apply generally to antibiotic resistance beyond the very nice, but potentially system-specific, results presented here; and 2) the terms themselves (history, chance, selection) could conceivably have different meanings in different contexts (more on this below).

Thanks, we better acknowledge these limitations. We have included a paragraph discussing the generality of our results in the discussion (Lines 437-450), and we have included a paragraph analyzing the assumptions done to analyze the forces (Lines 570-591). Also, see responses to #11, #12 and #13.

11) The study design has been used in numerous previous studies; it is well established, elegant, and has given rise to many new insights. However, as I understand it, there are some inherent assumptions of the approach that should be briefly discussed. Most notably, does the approach inherently assume that the effects of history, chance, and selection are additive (or perhaps linear) in some sense (in terms of phenotypic variance measurements or the Manhattan-based genotype metric)? While this simplifying assumption seems critical to the power of the approach, it is not clear to me that this assumption holds in general. When I try to think of this in terms of, say, a simple population dynamics model, the terms history, chance and selection are themselves somewhat nebulous, and it's not clear to me that they could be unambiguously and uniquely defined even in simplified theoretical models (or more directly, that the variance-based phenotype measures correspond to well-defined features or parameters of population dynamics models). I say this not to criticize the approach-again, its power lies in the simplicity of the design and the intuitive value of separating these three evolutionary features and attempting to quantify their contributions. But I think the article could be strengthened by briefly discussing the underlying assumptions-ideally by pointing to previous work (if it exists) that establishes that the features are additive and measurable in the sense required by the experimental design. If not, I think it would be worth discussing that limitation briefly, as I worry the inherently nonlinear nature of these very complicated, evolving systems could lead us to misinterpret the results. Given the general success of similar approaches in past work, I suspect the authors have thought through these issues in detail; discussing those points might open the paper to a broader audience not intimately familiar with all the previous studies.

Thank you for this thoughtful comment. To clarify, the history, chance, and adaptation study design itself does not make an assumption about the linearity or additivity of effects. The only linear assumption adopted in this study was through the use of a nested linear mixed model to estimate the forces in the phenotypic case. We opted to use such a model for its simplicity. Using more complex models that could account for non-linear effects, while being promising direction, was beyond the scope of the study.

Briefly, in the case of phenotypic variation, variation between populations is simply partitioned between three possible pools: (i) differences between ancestral and evolved populations (selection), (ii) differences between evolved populations with different ancestors (history) and (iii) differences between evolved populations with the same ancestor (chance). For genotypic variation, the same logic applies, using differences in the frequencies of mutations attributed to the same three pools. Where non-additive effects become important to consider is in interpreting the differences between the phenotypic and genotypic metrics, which we discuss further below (see #13). We added discussion of these assumptions in Lines 591-613.

12) One example related (but not identical to) the point above: in the current experiment, the role of history is defined in terms of previous selection conditions (drug and growth phase). But the new evolution experiment itself has multiple time points, and even qualitatively distinct epochs (sub- and super-MIC drug). So one might argue that history is playing a role continuously throughout the experiment-history not merely of the previous selection in fluoroquinolones, but also history of the previous time points / epochs of the new evolution (β-lactams). My point is that it is important to clearly define the terms at all stages and discuss, at least briefly, the limitations of the definitions that are chosen.

We agree that history is playing a continuous role during the experiment. According to Travisano, (Travisano et al., 1995) “the set of potential adaptations is severely limited to inherited constitution, so that at every moment the course of evolution is contingent on prior (historical) events”. We now acknowledge this more clearly that history plays a continuous role in lines 71-74 and 606-613.

For example, the mutations acquired at day 1 on the experiment arose by chance, were selected for in presence of CAZ or IMI, with a crucial importance of the previous evolutionary history, i.e. evolution in presence of CIP. If we jump, for example, to day 3, the previous mutations acquired on day 1 are right now historical mutations. We can apply this logic to any combination possible: mutations on day 2 are historical mutations if you analyze their evolutionary consequences at day 4, 5, 6… It just depends on the question that you want to answer. What we are asking here is how the previous exposure to an antibiotic X in lifestyles A and B, influences the further adaptation to an antibiotic Y. Therefore, we arbitrarily chose that everything happened before the exposure to the second antibiotic was just evolutionary history.

Analyzing the literature, most of the experiments following the Travisano design chose arbitrarily what they consider evolutionary history and from when they consider selection + chance (this even varies somewhat in the original article). Then, the roles of the forces are analyzed at the end of the experiment. However, there is an elegant paper from Rebolleda-Gomez and Travisano, where they analyze the contribution of the three forces periodically during their whole experimental evolution and not just at the end of the experiment (Rebolleda-Gomez and Travisano, Evolution 2019).

13) While history, chance, and selection are quantified at both the genetic and phenotype level, it's not clear to me that these numbers can be directly compared to one another (though it's tempting to do so!). Could the authors briefly comment on the connection between these measures-that is, when (and to what extent) one would expect correlations between them (e.g. high levels of historical influence at the genetic level leads to high levels of historical influence at the phenotype (MIC) level….assuming the definitions used here).

This is an interesting question. The contributions of history, chance, and selection at the genotypic and phenotypic levels would be same if every mutation that arose had an equal effect on the phenotype (or at least that the frequency of each mutation in the population was proportional to its phenotype) and phenotypic effects were additive, with no epistasis. The greater the deviation from those assumptions, the greater the differences will be between the genotypic and phenotypic roles of history, chance and selection. However, the differences between genotypic and phenotypic contributions may suggest additional information about the evolutionary system. In our results, the differences between the genotypic and phenotypic analyses were modest, suggesting relatively small deviations from these assumptions. The largest distinction was that in imipemen-selected populations, history contributed about 45% of genetic variation (Figure 4), but only about 33% of variation in imipenem resistance. This indicates that the mutations contributing to history, those in which parallel changes were observed in populations derived from the same ancestor, had a smaller effect on imipenem resistance than mutations associated with other categories. Interestingly, the “historical mutations” in these populations occurred most often in populations with ancestors previously evolved under biofilm conditions, implying constraining effects of this prior environment. It may be that some of these mutations were adaptations to the return to a planktonic environment, rather than to antibiotic selection. We further explain the connection between the phenotypic and the genotypic metrics in the methods (Lines 570-585).

14) Do the authors have any thoughts on how the results might be affected by the fact that the new evolution experiments take place in planktonic (rather than biofilm) conditions? How might the results differ if they had been performed in biofilm, and what could you learn from the fully symmetric experiment (P/ B initial strains evolved in both P and B new selection conditions). The authors may wish to discuss this avenue for future work.

Thanks for the comment. This is an interesting question, and we have asked this question ourselves too. We decided to run the experiment planktonically for one reason: after the CIP exposition, planktonic populations showed cross resistance to CAZ via mutations in adeJ and biofilm populations showed collateral sensitivity to CAZ via mutations in adeL. We believed that mutations in adeL had the potential to constrain evolution in CAZ while mutations in adeJ would be selected for. In fact, we detect mutations in adeB, which is regulated by adeJ.

We believe that the general take home message of the paper would be the same: selection is unsurprisingly the predominant force in the evolution of AMR and produced convergent evolution even at the nucleotide level in some instances. Yet history and chance play clear and important roles in the emergence of new resistance phenotypes, and possibly more so in biofilm. However, we argue that the reversions probably are lifestyle dependent, and therefore the likelihood of detecting the same reversions is low if we replayed the experiment in biofilm.

We have included this discussion in the manuscript (Lines 437-450).